# Identification Markers of Carotid Vulnerable Plaques: An Update

**DOI:** 10.3390/biom12091192

**Published:** 2022-08-28

**Authors:** Yilin Wang, Tao Wang, Yumin Luo, Liqun Jiao

**Affiliations:** 1Institute of Cerebrovascular Disease Research, Department of Neurology, Xuanwu Hospital of Capital Medical University, 45 Changchun Street, Beijing 100053, China; 2Department of Neurosurgery, Xuanwu Hospital of Capital Medical University, Beijing 100053, China; 3Beijing Geriatric Medical Research Center, Beijing Key Laboratory of Translational Medicine for Cerebrovascular Diseases, Beijing 100053, China; 4Beijing Institute for Brain Disorders, Capital Medical University, Beijing 100053, China

**Keywords:** carotid atherosclerosis, vulnerable plaques, stroke

## Abstract

Vulnerable plaques have been a hot topic in the field of stroke and carotid atherosclerosis. Currently, risk stratification and intervention of carotid plaques are guided by the degree of luminal stenosis. Recently, it has been recognized that the vulnerability of plaques may contribute to the risk of stroke. Some classical interventions, such as carotid endarterectomy, significantly reduce the risk of stroke in symptomatic patients with severe carotid stenosis, while for asymptomatic patients, clinically silent plaques with rupture tendency may expose them to the risk of cerebrovascular events. Early identification of vulnerable plaques contributes to lowering the risk of cerebrovascular events. Previously, the identification of vulnerable plaques was commonly based on imaging technologies at the macroscopic level. Recently, some microscopic molecules pertaining to vulnerable plaques have emerged, and could be potential biomarkers or therapeutic targets. This review aimed to update the previous summarization of vulnerable plaques and identify vulnerable plaques at the microscopic and macroscopic levels.

## 1. Introduction 

Worldwide, stroke contributes heavily to high mortality and disability rates [1]. The main pathological characteristic of stroke is atherosclerosis [2], which is a systemic, and inflammatory disease that is pathologically characterized by focal fibrosis, lipid accumulation and atherosclerotic plaques [2]. Commonly, based on stability, plaques can be classified into stable plaques and vulnerable plaques [2]. Vulnerable plaques are referred to as those that easily rupture and subsequently cause thrombosis and severe clinical events, such as stroke [3]. A higher vulnerability of plaques gives rise to a higher risk of stroke [4]. Atherosclerosis is frequently observed in large arteries, such as the carotid artery. Besides, special hemodynamics and large diameter contribute to the formation of vulnerable plaques in carotid arteries [4], which is a common cause of ischemic stroke [5]. Studies have pointed out that substenotic plaques may be considered as a cause of ischemic stroke and subsequent stroke recurrence [6]. Commonly, plaques are clinically silent, presenting asymptomatically. However, some vulnerable plaques may cause catastrophic ischemic stroke due to unexpected rupture, despite being in an asymptomatic state for decades. The morbidity of stroke in patients with asymptomatic carotid stenosis is approximately 0.5–1% every year. The potential mechanism for the shift from asymptomatic to symptomatic remains unclear [7,8]. Traditionally, risk stratification and therapeutic management of carotid atherosclerosis are mainly based on the severity of luminal stenosis. It has been gradually recognized that vulnerable plaques should also be taken into account [9]. At the microscopic level, vulnerable plaques are mainly characterized by lipid accumulation, macrophage and T lymphocyte accumulation, decreased vascular smooth muscle cells (VSMCs), extracellular matrix (ECM) degradation, etc [2,10,11]. At the macroscopic level, the classical features of vulnerable plaques include calcification, neovascularization, lipid-rich necrotic core (LRNC), intraplaque hemorrhage (IPH), thin fibrous caps, plaque surface ulceration, plaque rupture and plaque morphology [2,12,13,14]. Patients with vulnerable plaques are exposed to a high risk of ischemic stroke, even with mild (less than 50%) carotid stenosis [15]. Therefore, early identification of vulnerable plaques is conducive to better evaluation of risk and establishment of therapeutic intervention, preventing the severe outcomes caused by vulnerable plaques. This review aimed to provide a comprehensive review of the identification of carotid vulnerable plaques at both the microscopic and macroscopic levels. 

## 2. Identification of Vulnerable Plaques at the Microscopic Level

### 2.1. Inflammation 

C-reactive protein (CRP) is an inflammatory marker with no specificity that often accumulates in the macrophage-rich region of carotid plaques. Highly expressed CRP and high-sensitivity CRP (hs-CRP) indicate severe inflammation in plaques, implying plaque vulnerability [16]. Compared with asymptomatic patients, a high expression level of IL-6, IL -17A, IL-18, IL -21, and IL -23 in plaque has been detected in symptomatic patients with vulnerable plaques [17,18]. High expression levels of IL-6, IL-18 and IL-1β in carotid vulnerable plaques have also been proven in symptomatic patients with cardiovascular events [10,19]. Plaque T-cell gathering and highly expressed vascular cell adhesion molecule-1 have also been reported in symptomatic patients [17,18]. Tumor necrosis factor-α(TNF-α), a well-known inflammatory cytokine, contributes heavily to vascular endothelial cascade inflammatory reactions, promoting the progression and deterioration of atherosclerotic plaques [20]. Higher expression levels of TNF-α have been detected in patients with vulnerable plaques than in those with stable plaques [17]. In patients with frequent cardiovascular events, lower levels of interferon-γ (IFN-γ) and higher levels of macrophage inflammatory protein-1β (MIP-1β) and monocyte chemoattractant protein-1 (MCP-1) in carotid plaques have been reported, compared with those with fewer cardiovascular events [10]. In the late stage of differentiation, macrophages express a kind of inflammatory glycoprotein with no chitinase activity, namely YKL-40. Increased expression levels of serum YKL-40 indicate high plaque vulnerability. The potential mechanism may be related to the regulation of hyaluronic acid synthesis, MMP-9 activity. In addition, YKL-40 is implicated in cell migration, angiogenesis, and tissue remodeling [2]. Significantly higher soluble urokinase-type plasminogen activator receptor (suPAR) levels in plaque and plasma are observed in symptomatic patients than in asymptomatic patients. Plaque suPAR correlates with plaque inflammation [21]. As a Ca2+-binding protein, S100A12 (EN-RAGE, calgranulin C) belongs to the S100/calgranulins family and is mainly expressed in monocytes, neutrophils, and dendritic cells. Compared with healthy controls, plasma S100A12 is higher in patients with severe carotid stenosis, especially in those with symptoms in the last 2 months. IFN-γ and IL-1β can increase S100A12 levels [22]. 

### 2.2. ECM Degradation 

Matrix metalloproteinase (MMP) is a kind of protease generated by inflammatory cells. Therefore, MMP is a classical inflammatory biomarker that is a potential risk factor for plaque rupture [16]. The literature has proven that the high expression level of MMP-9 in plaques and serum indicates the instability and rupture tendency of plaques [23,24]. MMP-9 is highly expressed in the macrophage-gathered region of atherosclerotic plaques, which can degrade the vascular wall, ECM, and fibrous caps [2], and induce SMCs to migrate from the medial membrane into the intima [20]. In addition, MMP-9 can degrade collagen V, which contributes heavily to intactness and stability of basilar membrane and the fibrous caps, promoting plaque rupture [20]. In ApoE-/- mouse models, the natural herbal anti-inflammatory agents tanshinone IIA and astragaloside IV can decrease the instability of vulnerable plaques. The potential mechanism may be the inhibition of MMP-9 through the PI3K/Akt/TLR4/NF-κB pathway [25]. Additionally, vulnerable plaques have higher levels of MMP-1, MMP-2, MMP-7, MMP-8, MMP-12, and MMP-14, and lower levels of tissue inhibitor of metalloproteinases 3 (TIMP-3) has been observed in vulnerable plaques [16,17,24,26,27]. 

A Disintegrin and metalloproteinase with Thrombospondin motifs (ADAMTS) family can degrade ECM because of proteolytic activity, and includes 19 proteases. They participate in inflammation, coagulation, angiogenesis, and organ development [28]. A disintegrin and metalloproteinase with thrombospondin motifs type 4 (ADAMTS4) is a disintegrin and metalloproteinase with thrombospondin motifs 4 [29], which can be generated from macrophages, arterial SMCs, and endothelial cells [30,31]. Higher expression levels of ADAMTS4 in plaques and in the serum of patients with vulnerable plaques have been detected, compared with patients with stable plaques [30]. Immunohistochemical results show that ADAMTS4 is highly expressed in shoulder, fibrous caps, adjacent lipid core region, and macrophage-rich regions of carotid plaques [30]. Its expression level can be increased by some proinflammatory factors, such as IL-1, IL-6, TNF-α and leptin [30,32]. In vitro, ADAMTS4 knockdown has been proven to reduce macrophage infiltration into the ECM [33]. ADAMTS4 is equipped with platelet response elements and integrin structure domains, contributing to cell to matrix and cell to cell adhesion [30]. More importantly, ADAMTS4 can degrade versican, which is a component in fibrous cap ECM [34]. Versican, a chondroitin sulfate proteoglycan present, accumulates in plaque shoulders, fibrous caps and ECM, contributing to plaque integrity [30]. Compared with stable plaques, significantly decreased versican was detected in vulnerable plaques [30], which may be attributed to cell death, fibrosis and inflammatory events [30,35,36]. Genome-wide association studies point out that multiple single-nucleotide polymorphisms (SNPs) may participate in the potential mechanism of atherosclerosis [37,38]. ADAMTS7 is a zinc metalloproteinase [28]. rs3825807and rs7173743 of ADAMTS7 were associated with carotid plaque vulnerability. Specifically, the A/A genotype (A allele) of rs3825807 and the T/T genotype (T allele) of rs7173743 remarkably enhanced the vulnerability of plaques [28].

### 2.3. Lipid Metabolism 

Lectin-like oxidized low-density lipoprotein receptor-1 (LOX-1) is a scavenger receptor for low-density lipoprotein cholesterol (ox-LDL). LOX-1 can combine with ox-LDL in the arterial wall, leading to endothelial cell dysfunction and ECM degradation [39]. LOX-1 is qualified as an independent risk factor for carotid vulnerable plaques [2]. In pathological situations, extracellular domain of membrane bound LOX-1 is cleaved into a soluble form (sLOX-1), positively correlating with the expression level of LOX-1. The serum LOX-1 level can also be a potential biomarker for predicting plaque vulnerability [20,40]. Jens et al. established a single-domain antibody fragment (sdAb) that binds LOX-1 with high specificity, affinity, and fast blood clearance. LOX-sdAb is a promising probe to generate imaging tracers for identifying LOX-1 expression in plaques in vivo [41]. 

Nonhigh-density lipoprotein cholesterol (non-HDL-C) refers to the very low-density lipoprotein cholesterol, low-density lipoprotein cholesterol and intermediate density lipoprotein cholesterol [42]. Increased serum non-HDL-C levels (≥4.1 mmol/L) may indicate vulnerable carotid plaques in asymptomatic patients, and may be an independent risk factor for vulnerable plaques without interaction with age, sex, smoking, diabetes mellitus, or hypertension [42]. Studies have found that small dense LDL can activate inflammation related to peripheral blood mononuclear cells and endothelial cells [43]. In addition, dense LDL particles can affect plaque composition, especially the proportion of macrophages [44]. 

Previous studies reported that cholesterol crystals may be related to plaque vulnerability in coronary arteries [45]. Plaques with cholesterol crystals more frequently present large lipid arcs and thin-cap fibroatheroma than those without cholesterol crystals [46]. In carotid atherosclerosis, cholesterol crystals have also been proven to be an important element in vulnerable plaques. Plaques with cholesterol crystals are often accompanied by more macrophages and calcification, leading to more cerebrovascular symptoms [47]. Cholesterol crystals can activate the NLRP3 inflammasome and induce the secretion of IL-1 [48], IL-1β and C-reactive protein [49], activating local inflammation [47]. Via the complement system, cholesterol crystals can activate the inflammasome/caspase-1 and induce the release of cytokines (IL-1β and TNF) in whole blood [50], inducing systemic inflammation and vascular injury [47]. 

For patients with carotid stenosis ≥50%, higher triglycerides, and lower high-density lipoprotein cholesterol and tissue inhibitor of metalloproteinase 1 were detected in symptomatic patients than in asymptomatic patients [17]. For symptomatic patients, those with recent symptoms show higher lipoprotein-associated phospholipase A2 in blood than those with remote symptoms [51].

High serum free fatty acid (FFA) concentrations indicate high plaque vulnerability. FFAs are closely associated with lipid metabolism. Overly high serum FFA levels give rise to micelles and fatty acid vesicles with acidic cores fusing with endothelial cells, inducing plaques formation. Studies have suggested that high FFA levels are related to stroke events [52]. 

### 2.4. Cluster Differentiation Antigen and Chemokines

The monocyte count in vulnerable plaques is significantly higher than that in stable plaques [53]. CD163 is a unique surface protein of monocytes and macrophages [54]. CD163+ macrophages are often located in regions of IPH [55]. Higher CD163+ macrophages and CD163 mRNA levels are detected in vulnerable plaques than in stable plaques [56]. Circulating sCD163 level positively correlates to inflammation [54]. CD163 levels positively correlate with IL-6, IL-10, interleukin 1 receptor antagonist (IL-1RA), MIP-1β and MCP-1, and negatively correlate with type I cytokines, such as IFNγ and IL-12p70. CD163+ cells positively correlate with lipids, but negatively correlate with elastin, collagen, and SMCs [56]. 

For patients with high-grade internal carotid artery (ICA) stenosis, a higher level of plasma sCD36 is detected in those with recent symptoms than in those with remote symptoms. In plaques, CD36 is often present in intima with macrophage-gathering [57]. 

In carotid plaques, CD146 is mainly located in infiltrated macrophages and blood vessels, and significantly correlated with MMP-9. Intraplaque CD146 positively correlates with serum sCD146. The serum sCD146 level is associated with IL-6, MMP-9, and hsCRP [58]. 

COMP (cartilage oligomeric matrix protein), expressed by VSMCs, can interact with collagen I and growth factors [59], promote collagen fibrillogenesis and maintain the contractile phenotype of VSMCs, regulating plaque vulnerability [60]. COMP increases in vulnerable plaques, showing a positive correlation with CD68- positive and lipid- positive areas, and a negative correlation with elastin, collagen, and SMCs. Notably, COMP colocalizes with CD163 cells, suggesting that COMP may modulate the macrophage phenotype [60]. Similarly, CD163 cells also increase in vulnerable plaques, relating to inflammation, angiogenesis and vascular permeability [60]. Altogether, COMP may decrease plaque stability by regulating the function and polarization of CD163 macrophages [61]. 

Src homology 2 (SH2) domain-containing protein tyrosine phosphatase 1 (SHP-1) is a classical nonreceptor protein tyrosine phosphatase. Immunohistochemical staining results show that SHP-1 shares similar distribution with CD68, such as in fibrous caps, plaque shoulders and necrotic areas. The expression of SHP-1 increases with the progression of carotid plaques. The increased protein expression of SHP-1 indicates a high plaque vulnerability. The underlying mechanism may be associated with the regulation of efferocytosis mediated by macrophage polarization [62]. 

Serum macrophage CXC-chemokine ligand 16 (CXCL16) is a potential biomarker for carotid vulnerable plaques. Significantly higher serum CXCL16 levels were detected in vulnerable plaques and the micro-embolic signals (MES) positive group than in the stable plaque- and MES-negative groups. Serum CXCL16 levels rise with plaque area, lumen stenosis rate and intima-media thickness [63]. 

Significantly increased expression levels of chemokine (c-c-motif) ligand 19 (CCL19) have been detected in vulnerable plaques. In addition, CCL19 often colocalizes with CD3+ T-cell lymphocytes. CCL19 may be a predictor for vulnerable plaques [64]. 

### 2.5. MiRNA

MiRNA expression analysis carried out in atherosclerotic plaques has proven the important role that miRNAs play in vulnerable plaques [65]. Higher expression levels of miR-100, miR-125a, miR-127, miR-133a, miR-145, and miR-221 were detected in symptomatic plaques than in asymptomatic plaques. Intraplaque miR-125a negatively correlates with the serum low-density lipoprotein cholesterol [66]. In another study, no significant difference between the expression levels of miR-100 and miR-127 were observed, which is inconsistent with previous results. The expression level of miR-133a, miR-145, and miR-221 in VSMCs may correlate to the differentiation and migration of VSMCs. In addition, miR-133a may regulate MMP-9 and the plasminogen activator inhibitor PAI1, which are involved in the regulation of plaque vulnerability. MiR-221 may regulate inflammation, proliferation, and angiogenesis. MiR-145 may regulate cholesterol release from cells [67]. 

A receptor of granulocyte macrophage colony-stimulating factor (GM-CSF), known as CSF-2R, includes two subunits (CSF-2Rα and CSF-2Rβ) [68]. High expression levels of CSF-2Rα have been detected in macrophage-rich regions of vulnerable plaques [3]. Macrophages contribute heavily to the progression of atherosclerosis [69]. GM-CSF/CSF-2Rα signaling can induce an atherogenic inflammatory response via the JAK2/STAT5 pathway [68]. Subsequently, miR-532-3p was identified to combine with the 30 UTR of CSF-2Rα to downregulate CSF-2Rα. Accordingly, an abnormal miR-532-3p- CSF-2Rα axis is thought to contribute to the development of vulnerable plaques. The potential mechanism may be related to low-density lipoprotein (LDL) or TNF-α [3]. Recently, dysfunction of the miR-532-3p- CSF-2Rα axis has been verified in patients with vulnerable plaques and Apoe/mouse plaques. In addition, agomiR-532-3p therapy has been proven to inhibit the expression of macrophage CSF-2Rα, successfully stabilizing the vulnerable plaques [3]. 

The serum expression levels of miR-124, IL-1β and TNF-α in vulnerable plaques were significantly increased compared with those in stable plaques. In addition, the serum expression level of miR-124 positively correlates with IL-1β and TNF-α [70]. The three may be potential biomarkers for the early identification of vulnerable plaques in patients at risk of acute cerebral infarction. The combination of the three yielded the better diagnostic efficacy, with an AUC of 0.853 (95% CI: 0.790-0.915), a sensitivity of 82.80%, and a specificity of 78.90% [70].

Transforming growth factor-beta (TGF-β) can regulate the proliferation and migration of cells, calcification and matrix synthesis [71]. DACH1 (dachshund homolog 1) suppresses apoptosis and the TGF-β signaling pathway [72]. Accordingly, DACH1 may be involved in atherosclerosis. In tumors, miR-484 can regulate angiogenesis and promote necrosis [73]. Recently, the potential role that miR-484 play in atherosclerosis has been identified. An interaction between miR-484 and DACH1 was found, which should be further explored in atherosclerosis and vulnerable plaques [8]. 

G-protein-coupled receptor 56 (GPR56) shows significantly differential expression levels between carotid stable plaques and vulnerable plaques in the carotid [8]. GPR56 can suppress vascular endothelial growth factor and angiogenesis and is involved in vascular remodeling. In addition, GPR56 is implicated in ECM remodeling [74], and also participates in cell adhesion [75]. MiR-942 is negatively correlated with GPR56. Because it is regulated by AKT, miR-942 can inhibit apoptosis and influence carotid plaque vulnerability [8]. 

Differential expression of APoD has been found between stable plaques and vulnerable plaques in the carotid. ApoD participates in reverse cholesterol transportation and inhibits proliferation of VSMCs in culture [8]. In addition, the antioxidative effect of ApoD should also be given enough attention [76]. Accordingly, ApoD may be involved in the regulation of plaque vulnerability. MiR-214 has been detected in the exosomes secreted by human microvascular endothelial cells. MiR-214 can induce cell migration and angiogenesis, as well as inhibit senescence [77], and miR-214-3P is negatively correlated with APOD and is a potential regulator of plaque vulnerability. The potential mechanism needs further verification [8]. 

### 2.6. Others 

Baculoviral IAP Repeat Containing 6 (BIRC6) is a ubiquitin-conjugating E2 enzyme that negatively modulates apoptosis and autophagy. BIRC6 can modulate the fusion of autophagosomes and lysosomes, inhibiting autophagy [78]. In addition, BIRC6 inhibits autophagy by regulating MAP1LC3B, which is involved in autophagosome synthesis and the selection of autophagy substrates [79]. LC3B is the product of the MAP1LC3B gene and is related to proteasomal degradation and ubiquitination [80]. Ubiquitination serves as an important regulatory mechanism in autophagy [78]. In advanced carotid plaques, macrophages, ECs and SMCs present activated autophagy, which may be related to plaque vulnerability [81]. Remarkably higher BIRC6 mRNA levels were found in symptomatic patients than in those without symptoms and were especially higher in those with the rs35286811 risk allele [12]. The results of genotype comparisons show that an exonic SNP of BIRC6, named rs35286811, is related to cerebrovascular symptoms [12]. Significantly decreased MAP1LC3B mRNA levels were detected in patients with vulnerable plaques. BIRC6 may give rise to ubiquitination and degradation of MAP1LC3B, downregulating autophagy and regulating plaque vulnerability [12]. 

Serum osteoprotegerin [82], and pregnancy-associated protein A, are significantly higher in patients with vulnerable plaques than in those with stable plaques [17]. In addition, the concentrations of osteopontin and osteoprotegerin in plaques in patients with recent neurological symptoms are obviously higher than those in asymptomatic patients [17,83].

Osteopontin targeted theranostic nanoprobes have been proved to precisely regress vulnerable atherosclerotic plaques with the guidance of fluorescence/MR imaging [84]. Cleaved osteoglycin is significantly higher in asymptomatic plaques than that in symptomatic plaques, correlating to the histological vulnerability index of plaques. The potential mechanism may be related to reducing cell apoptosis and retaining low-density lipoprotein [85]. 

Insulin can decrease plaque vulnerability via nitric oxide synthase (NOS)-dependent mechanisms [86]. Notably, a correlation between fasting glucose levels and plaque vulnerability was not observed [87]. Insulin increases the expression of vascular endothelial growth factor [88], leading to abnormal angiogenesis with hemorrhage and leakage tendency. In mouse atherosclerosis models, insulin enhances SMCs and collagen, and decreases plaque necrosis and macrophage accumulation, decreasing plaque vulnerability [87]. 

Fibrinogen belongs to the hemostatic system. The fibrinogen γ′ is a variant of fibrinogen. Fibrinogen γ′/total fibrinogen ratio is related to ischemic stroke [89]. In recently symptomatic patients with mild-to-moderate carotid stenosis, fibrinogen and fibrinogen γ’ negatively correlate with the volume of IPH and LRNC, which are characteristics of vulnerable plaques. This correlation seems to be irrelevant to inflammation, and indicates that fibrinogen and fibrinogen γ’ are negatively regulated to the development of vulnerable plaques [90].

Von Willebrand Factor (VWF) can induce platelet adhesion and aggregation. ADAMTS13 can cleave ultralarge VWF multimers into smaller sizes. Both have been proven to be related to cerebral infarction [91]. Other studies verified that the VWF antigen level and ADAMTS13 activity seemed to be irrelevant to classical characteristics of vulnerable plaques, such as the volume of IPH and LRNC, and plaque ulceration [90]. 

Based on endothelial expression, increased expression levels of vascular cell adhesion protein-1(VCAM-1) have been detected in vulnerable plaques, compared with stable plaques. However, vWF, P-selectin, and LOX-1 do not show the potential for predicting vulnerable plaques, although they contribute heavily to plaque biology [92]. 

Human cytomegalovirus (CMV) infection can lead to increased proliferation and impaired apoptosis. Persistent infection interrupts the normal function of endothelial cells (ECs) and activates the proinflammatory pathway related to specificity protein 1, nuclear factor κB, phosphatidylinositol 3-kinase, and platelet-derived growth factor receptor, promoting monocyte and SMC proliferation and migration into the vascular intima, as well as lipid accumulation and expansion within lesions [93]. Intraplaque CMV levels are related to lymphocyte immune-activation and hs-CRP [94]. In patients with carotid atherosclerosis, anti-HCMV antibody levels are positively correlated with IMT and TNF-α [95]. A study proved that CMV infection can increase the serum levels of TNF-α, LOX-1 and MMP-9, increasing plaque vulnerability [20]. 

BCLAF1 (BCL2 (B-cell lymphoma 2)-associated transcription factor 1) participates in the transdifferentiation of SMC into a macrophage-like phenotype, serving as a potential marker of carotid vulnerable plaques [96]. 

The transcription factor interferon regulatory factor-5 (IRF5) promotes macrophages to present a pro-inflammatory state, driving the formation and rupture of carotid vulnerable plaques in mouse carotid plaque rupture model [97]. 

VSMCs are one of the major sources of CTH (cystathionine gamma-lyase)-hydrogen sulfide (H2S). Intraplaque CTH level positively correlates with collagen content and negatively correlates with CD68+ and necrotic core area. CTH- H2S attenuates atherosclerotic plaque vulnerability via TFEB (transcription factor EB)-mediated autophagy, serving as a biomarker of vulnerable plaques [98]. 

Compared with carotid asymptomatic plaques, 2.2-fold upregulation of glutamine synthetase (GLUL) mRNA has been found in stroke-causing plaques. In the post-symptomatic period, a declined mRNA expression level of GLUL has been observed, indicating that GLUL may participate in plaque destabilization and rupture [99]. 

The microscopic factors that potentially predict vulnerable plaques are summarized in Table 1. In Table 1, some factors related to the mechanism of plaque progression (such as lipid metabolism and inflammation) may play roles in the whole process of plaque progression. Accordingly, they may be detected in all plaques, and they are factors for plaques. However, the expression levels of these factors in stable plaques and vulnerable plaques show significant differences, which also may be valuable for identifying vulnerable plaques. In other words, their existence may be not specific to vulnerable plaques, while the significantly differential expression levels may be specific to vulnerable plaques, and they may be potential markers of vulnerable plaques.

## 3. Identification of Vulnerable Plaques at the Macroscopic Level

### 3.1. Ultrasound (US)

Ultrasound (US) is widely used in carotid atherosclerosis because of the features of fast examination, no radiation exposure [100], low cost, availability, simplicity, and noninvasiveness. US can identify stenosis scope and degree, as well as plaque shape and properties [52]. In the early period, plaque composition was evaluated by US based on visual assessment of echogenicity and heterogeneity: type I: uniformly echolucent; type II: predominantly echolucent (>50%); type III: predominantly echolucent (<50%); type IV: uniformly echogenic; type V: highly calcified plaques or unclassifiable acoustic shadow [101]. Recently, many US methods have been established to evaluate features of plaque vulnerability, such as the assessment of intima media thickness (IMT), pulse wave velocity (PWV), and grayscale median (GSM) [102]. IMT is widely used to assess carotid plaques with relatively low specificity [102]. PWV can suggest the stiffness of detected arterial wall segments via noninvasive pulse wave imaging [103]. Currently, GSM is most widely used to evaluate plaque vulnerability [100]. Computer-assisted techniques have contributed to the standardization of grayscale methods. For example, Adobe Photoshop can calculate the GSM value of plaques easily and reproducibly by analyzing the grayscale histogram of plaques [104]. A grayscale median (GSM) value displays the distribution of pixel brightness [105]. Images are normalized with vessel walls (adventitia) and blood as the reference for white and black, respectively. Transforming histology simples into pixel brightness values can identify the composition of plaques and predict plaque vulnerability [106]. In patients with ICA stenosis > 50%, lower GSM values are observed in symptomatic patients than in asymptomatic patients [17]. Low GSM values can predict vulnerable plaques [107]. Previous studies have found that a median GSM of 25.5 was related to cerebral infarction. A GSM of 35 may be an appropriate cutoff point for evaluating plaque vulnerability [101]. Another study suggests that GSM < 25 can indicate a vulnerable plaque [108]. The cutoff point for identifying vulnerable plaques remains contradictory. The literature has pointed out that lower GSM values may be related to larger lipid cores, while higher GSM values are related to more calcium and fibrous content in plaques [106]. Another study suggested that plaque echogenicity seems to be irrelevant to intraplaque neovascularization [104]. Additionally, several special features related to vulnerable plaques have been observed. A large plaque area (>95 mm^2^) significantly correlates with histologically vulnerable plaques [108]. The discrete white area (DWA) is referred to as a noncalcified area without acoustic shadow in black areas [109]. Usually, DWAs are hyperperfused and related to neovascularization and increased macrophages [109]. In US imaging, plaque ulcer is defined as an intraplaque concavity with a stronger echo on the adjacent plaque surface and a relatively weaker echo on the basal border [110]. The juxtaluminal black (hypoechoic) area (JBA) is known as an area of pixels next to the lumen with a low grayscale value (<25) and without echogenic caps [111]. A large JBA (>6 mm^2^) may indicate vulnerable plaques [108]. A large JBA relates to a higher ulceration score, which is a classical feature of vulnerable plaques [106]. 

Three-dimensional ultrasound (3D US) is an accurate and effective tool to evaluate the composition, volume, and morphology of plaques, even to monitor therapeutic effects on carotid plaques, improving the quantification and visualization of carotid plaques [112]. In clinical practice, 3D US can assess the volume, wall and thickness of plaques, blood flow, and morphological features of carotid [113]. In 3D-US, vascular plaque quantification is capable to obtain real-time 3D volume imaging and quantitatively analyze components and morphology of plaques. Compared with the low-risk group, the medium-low-risk group, medium-high-risk group, GSM is the lowest in the high-risk group. Additionally, GSM negatively correlates with lipid core ratio and may predict the tendency of plaque rupture, qualified to assess the plaque vulnerability [114]. 

Ultrafast ultrasound imaging (UUI) can assess stiffness and the distribution of stiffness inside plaques via shear wave elastography (SWE). SWE results show that stiffness range of 3–5 m/s is frequently observed in vulnerable plaques [115]. 

Contrast-enhanced US (CEUS) is a novel and new noninvasive imaging technique [116] that can clearly display vascular dimensional position, especially intraplaque neovascularization [116]. Other than CEUS, high-resolution MRI can also assess neovascularization. Compared with high-resolution MRI, CEUS is more available, cheaper, faster, and more compatible with implants such as pacemakers [117]. Common CEUS enhancement was divided into four grades. Grade I: no enhancement; Grade II: vasa vasorum enhancement in adventitia or periadventitial tissue; Grade III: intraplaque neovascularization enhancement on adventitial side or shoulder of plaques. Grade IV: widespread plaque core enhancement [101]. CEUS enhancement implies high vascular density (neovascularization) and destroyed vascular integrity (IPH), which are classical characteristics of vulnerable plaques [16]. Contrast enhancement with high grade and intensity suggests high microvascular density, indicating a high incidence of stroke [101]. Increased density and number of microvascular networks in advanced plaques are observed. These fragile and leaky microvascular networks lead to IPH and increased inflammation [118]. Other than number, morphology and maturity of intraplaque microvessels may also contribute to plaque vulnerability. Symptomatic plaques have immature and dysmorphic microvessels, characterized by dilation, multilobules and the absence of SMC. In addition, in the surrounding area of such vessels, vascular endothelial growth factor colocalizing with macrophages was detected. Such vessels may induce inflammatory cell recruitment and serve as vascular leakage sites, increasing plaque vulnerability [119]. CEUS can assess intraplaque neovascularization (IPN), which can be divided into three grades: grade 0: no enhanced microbubbles; grade 1: moderate enhanced microbubbles restricted to the shoulder and/or adventitial side of plaques; grade 2: extensive enhanced microbubbles restricted to the core of plaques. Grade 2 IPN detected by CEUS indicates the risk of ischemic events in asymptomatic patients [120]. In addition, CEUS enhancement may be related to serum inflammatory biomarkers [16], reflecting intraplaque inflammation and having the potential to be a tissue-specific marker of inflammation [121]. Compared with asymptomatic plaques, symptomatic plaques identified by late- phase CEUS show higher levels of CD68, IL-6, CD31, MMP-1 and MMP-3, indicating more inflammation, angiogenesis and matrix degradation [122]. Kim et al. found that intraplaque neovascularization identified by CEUS was associated with serum MMP-9 levels but irrelevant to hs-CRP [123]. Another study suggested that CEUS reveals blood flow rather than clear vascular anatomy, which indicates that contrast enhancement cannot directly reflect inflammation, but rather neovascularization, the outcome of inflammation [124]. Additionally, the literature has pointed out that neovascularization seems to be not systematically associated with inflammation. The two processes do not seem to occur simultaneously, and a temporal interval has been observed between the two processes. Accordingly, compared with neovascularization, inflammation may be more sensitive in predicting the occurrence of symptoms [125]. 

Some limitations of CEUS should be considered. In patients with carotid atherosclerosis, plaque neovascularization on CEUS may indicate the possibility of stroke recurrence. Notably, in plaques with severe calcification, CEUS may be less reliable [117]. In addition, some doubts have been raised regarding the relationship between CEUS enhancement and neovascularization density. The literature found that compared with histologically confirmed nonvulnerable plaques, higher neovascularization density has been observed in histologically confirmed vulnerable plaques, while the discrepancy in contrast enhancement between the two types of plaques was not significant. This indicates that further studies are warranted to explore whether CEUS enhancement can be a potential predictor of vulnerable plaques [124]. In addition, CEUS cannot assess IPH accurately, accordingly, the possible confusion of IPH and neovascularization may be considered when explaining the results [126]. Regarding ulceration identification, CEUS is more sensitive and more accurate than color Doppler ultrasound (CDUS) [127]. For CEUS, ulceration is defined as a microbubbles column ≥ 1 × 1 mm within a plaque [16]. The better diagnostic power of CEUS depends on the use of microbubble contrast agents. Even in the identification of small ulcerations, CEUS can be more sensitive than CTA [128]. However, the danger carried by the microbubble contrast agents must be considered, such as microembolism, toxicity, and inertial cavitation [128].

### 3.2. Magnetic Resonance Imaging (MRI) 

MRI may be a relatively accurate method to evaluate IPH and stratify the risk of cerebrovascular events [126]. Intraplaque IPH commonly presents hyperintense signals on T1-weighted (T1-W) and time-of-flight (TOF) sequences, while it presents variable signals on T2-weighted (T2-W) and proton density sequences [126,129]. IPH is commonly found ipsilateral to embolic stroke of unclear source [26]. For symptomatic patients, IPH identified on MRI may predict ipsilateral ischemic events [126]. For asymptomatic patients, the T1-W signal in plaques can also predict ischemic events [130]. MRI studies have proven that IPH relates to plaque enlargement and small leaky neovessels, causing IPH, plaque growth and vulnerability [124]. Cerebral microbleeds (CMBs) are small, rounded low intensity areas and are usually observed on T2-weighted MRI [131]. Patients with CMBs often show more vulnerable plaques [131] and higher inflammatory marker levels than those without CMBs [132]. A previous study proved that CMBs correlate with fatty plaques and clinical symptoms in patients with carotid atherosclerosis [133]. TOF is often used in noncontrast-enhanced MRA, which can present vascular contrast via a high signal of moving blood. Randomly changing the image direction is a unique advantage. However, TOF-MRA shows poor accuracy in ulceration detection, which may be attributed to complicated influencing factors (shape, location and orientation of ulceration) [128]. Multi-contrast cardiovascular magnetic resonance (CMR) vessel wall imaging can noninvasively identify plaque elements, consisting of T1-W, T2-W and TOF sequences. Significantly discrepant T1 values have been found between IPH, NC and the loose matrix. More recent hemorrhage may present lower T1 values, while loose matrix often displays long T1 values. NC showed extensive T1 values due to the multiple elements (cholesterol crystals, apoptotic cells and calcium particles). Accordingly, T1 values are well qualified for identifying IPH, even the IPH phase. However, differentiating loose matrix and NC solely depends on T1 values and may not be reliable [134]. Traditionally MRI often identifies plaque components depending on a combination of series sequences, such as T1-W imaging, T2-W imaging and TOF. The potential limitations may be considered, including limited spatial coverage and slice resolution, long scan time, and even underlying misregistration between different images [135]. 

Multicontrast high-resolution-MRI is a noninvasive technique with high spatial resolution that can detect and quantify the location, morphology and components of carotid plaques with adequate safety and high accuracy, sensitivity and specificity [26,136,137]. High-resolution MRI can identify several classical features of vulnerable plaques, such as calcification, IPH, inflammatory tissues, thin and ruptured caps and LRNC [138,139]. Accordingly, the identified features may provide guidance for risk stratification of carotid plaques [140]. Contrast-enhanced MRA (CE-MRA) can present better vascular imaging via paramagnetic contrast agents [128]. Gadolinium and iron particles are common contrast agents [129]. Iron oxide particles can enter plaques and gather in macrophages within inflamed vascular walls, and are thus qualified for assessing plaque inflammation. The focal area with accumulation of iron oxide often presents signal absence, indicating vulnerable plaques [129]. Plaque enhancement after gadolinium administration on CE-MRA may predict neovascularization [129]. In addition, CE-MRA can identify more ulcerations than TOF-MRA, especially ulcerations in calcified plaques. However, the high price and gadolinium toxicity of CE-MRI must be considered [128]. Even in patients without renal impairment, gadolinium accumulation has been found in various tissues, such as bone, brain, and kidneys [141,142,143]. Blood suppression in MRI imaging can give rise to better contrast between blood and tissue, known as black-blood MRI (BB-MRI) [102]. BB-MRI can better evaluate the lumen area and wall thickness of vessels, as well as the total vascular area [144]. A study has proven that a high T1-W signal on BB-MRI predicts vulnerable plaques [145]. BB-MRI evaluates plaques depending on the strength ratio with surrounding tissues of plaques. Stable comparison with the tissues is vital for BB-MRI [146]. A novel sequence named multicontrast atherosclerosis characterization (MATCH) has been established, which can measure three different contrast weightings (T1-W, T2-W, and gray blood) simultaneously via a 3D fast low-angle shot technique in a shorter scan time. A study has proved that compared with traditional multicontrast sequences, MATCH can identify more calcifications with a sensitivity of 100.0%. For other features such as IPH, LRNC and the loose matrix, the two identification methods show the same accuracy [135]. For identifying thin fibrous caps, MATCH may not be reliable. In addition, relatively low slice resolution may be considered in clinical practice [147]. A new MRI sequence, named simultaneous noncontrast angiography and intraplaque hemorrhage (SNAP) is characterized by generating gray blood reference (Ref), black blood corrected real (CR), bright blood MR angiography (MRA) image sets, which can detect ulceration or stenosis colocalizing with IPH with high scan efficiency. A study verified the consistent accuracy between conventional multi-contrast sequences and SNAP sequences in the detection of juxtaluminal calcification. In addition, SNAP sequences can identify more ulcerations than conventional methods with high sensitivity [148]. However, SNAP identifies chronic or old hemorrhages with low sensitivity. Besides, poor differentiation between IPH and lipid pools or loose matrix should be considered [149]. 

### 3.3. Positron Emission Tomography (PET)

Further, 18F-Fluorodeoxyglucose (18F-FDG) is a common tracer of positron emission tomography (PET) examinations, which can noninvasively indicate intraplaque inflammation and glucose metabolism [150]. Thus, 18F-FDG can be absorbed by intraplaque activated macrophages because of high glucose metabolism in macrophages [150]. In addition, intraplaque inflammation relates to vascular endothelial growth factor and neovascularization, which is also a feature of vulnerable plaques [117]. Accordingly, high 18F-FDG uptake may predict intraplaque neovascularization [126]. Further, 18FDG-PET can also identify lipid-rich plaques with high sensitivity and specificity. The results of 18FDG-PET may be related to CD68 and MMP-9 [17,151]. Higher 18FDG uptake has been observed in symptomatic patients, especially those with cerebrovascular events within 3 months. Moreover, 18F-FDG uptake may predict the activity and vulnerability of plaques [26]. For patients with moderate and severe stenosis, plaque FDG uptake relates to recurrent stroke. Higher FDG uptake may predict vulnerable plaques [150]. 64Cu-labeled divalent (containing two RGD motifs) cystine knot peptide, known as 64Cu-NOTA-3-4A, can bind with intraplaque αvβ3 with high affinity and specificity. Integrin αvβ3 has been proven to be a biomarker of vulnerable plaques and is highly expressed by macrophages, endothelial cells, and SMCs. Accordingly, 64Cu-NOTA-3-4A may serve as a novel PET tracer for identifying vulnerable plaques [152]. High cost and ionizing radiation are two main disadvantages of PET scans [153]. 

### 3.4. Computed Tomography Angiography (CTA) 

Computed tomography angiography (CTA) can evaluate plaque morphology and components, as well as the lumen situation [154]. Combined with a semiautomatic image analysis procedure, CTA can well describe general parameters (total volume, diameter and area of stenosis) and classical features of vulnerable plaques (LRNC, IPH, and calcium), thus identifying vulnerable plaques and even quantifying plaque progression or rehabilitation [155]. CTA can better describe lumen morphology and ulcerations than US. For some small ulcerations, CTA may not be that reliable [129]. Histopathology evidence has proven that thin fibrous caps and fissured fibrous caps are related to moderate and high risks of plaque rupture, respectively [26]. Fissured fibrous caps may present a more significant enhancement on CTA imaging than nonfissured caps [156]. Similarly, another study also strengthened the result that fibrous cap rupture relates to postcontrast enhancement on CTA imaging [129]. Given the wide overlapping Hounsfield units (HU) value between IPH and LRNC, some studies suggests that CTA may be unreliable for differentiating IPH and LRNC, especially in cases of small ulcerations [26]. Other work also points out that IPH and connective tissues share similar densities, increasing the difficulty in discriminating between the two [126]. Accordingly, identifying IPH or LRNC via CTA remains a challenge [129]. While the latest literature points out that low densities (<25 HU) on CTA seem to indicate IPH. Compared with asymptomatic plaques, HU values of <25 are frequently observed in symptomatic plaques [157]. CTA can also identify and quantify neovascularization. Neovascularization seems to be positively corelated with the contrast enhancement amount on CTA imaging [26]. CTA can also identify the volume and thickness of plaques as well as vascular remodeling [26,129]. The disadvantages of CTA are as follows: (1) risk related to radiation exposure; (2) side effects caused by contrast material, such as anaphylactic reaction and contrast-induced nephropathy; and (3) the limited fatty tissue contrast [129]. Recently, multidetector CT scanners (MDCT) and volumetric analysis software have emerged to increase the accuracy and efficiency in capturing vascular features with decreased discomfort [26]. The combination of the modern MDCTA and analysis software can identify artifacts related to calcification, accurately differentiate IPH and LRNC, and avoid radiation and side effects related to contrast agents, overcoming the previous limitation of CTA. Moreover, MDCTA even identifies ulcerations with higher sensitivity and specificity, compared with DSA [26]. 

### 3.5. Digital Subtraction Angiography (DSA)

Digital subtraction angiography (DSA) is considered as the gold standard for diagnosing carotid atherosclerosis [2]. However, DSA cannot give definite diagnosis and classification until the late stage of luminal stenosis. In other words, DSA shows poor capacity to identify vulnerable plaques in the early stage [26]. In addition, due to its invasiveness, radiation and high cost, DSA is not popular in clinical practice [52]. 

### 3.6. Optical Coherence Tomography (OCT)

Optical coherency tomography (OCT) can present high-resolution images via reflection of near-infrared light. High resolution OCT can identify plaque rupture or erosion, thickness and the macrophage content in fibrous caps, as well as subintimal lipid accumulation [158]. Because of its invasiveness, OCT is not widely used in clinical practice. To date, OCT is mainly used in coronary atherosclerosis, while research has shown that it also has the potential for evaluating carotid plaques [26]. Table 2 displays the imaging features of vulnerable plaques, as well as advantages and disadvantages of imaging methods to identify vulnerable plaques. 

For patients with carotid plaques, imaging methods are necessary and convenient to evaluate the vulnerability of plaques. To date, various imaging methods have been proved to identify the vulnerable plaques. Vulnerable plaques consist of many types, such as IPH, neovascularization, and LRNC. Notably, each method has limitations and strengths, and solely one imaging method cannot cover all types of vulnerable plaques. In the future clinical practice, plaques should be evaluated by many imaging methods simultaneously based on individual situation. Combined with many imaging results, the evaluation of plaques may be reliable. 

### 3.7. Geometry and Morphology of Arteries 

In the early stage of plaque formation, thickened vascular wall and lumen stenosis can be compensated by arterial outer wall expansion, preserving the luminal area and blood flow [144]. Expansive remodeling (ER) is a common morphological change in the carotid artery [159]. While the compensatory capacity of the vascular wall is limited, ICA can be compromised under the condition that the maximum wall thickness is less than 1.5 mm. Otherwise, greater thickness leads to smaller lumens [160]. Previous studies have suggested that ER seems to be irrelevant to stroke symptoms and significant luminal stenosis, and thus may not be a reliable predictor for vulnerable plaques [161]. Other literature proved that carotid artery morphology may be related to plaque stability. In long-axis high-resolution MR images, the ER ratio refers to the maximum distance between the lumen and outer boundary of plaques perpendicular to the internal carotid artery (ICA) axis/the maximum luminal diameter of the non-arteriosclerotic distal ICA. The ER ratio is closely related to cerebral ischemic symptoms and may be a potential marker of vulnerable plaques [162]. Carotid artery geometry is accessed based on ICA angle and external carotid artery (ECA) angle, which are defined as the angles between the common carotid artery (CCA) and ICA and ECA, respectively. Compared with asymptomatic patients, greater ER ratio was frequently observed in symptomatic patients. In addition, ER relates to ICA angle. Compared with patients with slight ER, significantly larger ICA angle is observed in those with extensive ER. The ECA angle seems to be irrelevant to ER [159]. Besides, expansive arterial remodeling can also predict the incidence of ischemic complications in carotid artery stenting [163]. The combination of ER, characteristics and plaque morphology may be conducive for risk stratification [164]. Despite ER of the vascular wall, wall thickness can influence the plaque vulnerability. Maximum wall thickness is defined as the maximum distance between the lumen and vascular walls, which can be obtained via carotid MRI and is a marker for vulnerable plaques. Compared with lumen stenosis, maximum wall thickness can indicate vulnerable plaques with higher accuracy [165]. 

Carotid atherosclerotic arteries show smaller artery tortuosity and bifurcation angles than healthy carotid arteries. In addition, tortuosity, bulb diameter, and bifurcation angle increase along with age, which may be attributed to fragmentation and degradation of elastin [166]. Carotid bifurcation geometry can affect hemodynamics and early wall thickening [167] and has been identified as a potential risk factor for vulnerable plaques. The results of multi-contrast MR vessel wall imaging show that vulnerable plaques are commonly located near the flow divider level [168], especially for vulnerable plaques with IPH [137], while stable plaques often display distal locations [168]. In addition, carotid vulnerable plaques are often accompanied by smaller lumen expansion at the bifurcation, even in those with mild stenosis or non-stenosis. Currently, geometric features can be easily observed via angiography imaging, which may be conducive for clinical practice. Further studies are needed to verify the results via 4D flow MR imaging or computational fluid dynamics simulation [168]. 

In the cerebrovascular system, a circle of Willis (COW) is vital for hemodynamic regulation [169]. A previous study suggested that for patients with carotid artery disease, the difference in COW anomalies in the symptomatic group and asymptomatic group seems to be nonsignificant [170]. Another study suggests that in carotid atherosclerosis, hemodynamics contribute to plaque vulnerability. An incomplete COW is closely related to IPH. For other classical features of vulnerable plaques, such as FCR and LRNC, no significant correlation between them and incomplete COW is observed [171].

### 3.8. Geometry and Morphology of Plaque

The morpho-mechanical features of plaques can influence the shear stress, relating to the rupture tendency. Commonly, the proximal end of the stenosis suffers from higher shear stress than the distal part [14]. Carotid plaques can be divided into three types based on longitudinally symmetrical characteristics of the plaque shape. Type-I plaques are characterized by greater arc-length located in the downstream vascular wall above the position of maximal wall thickness (WTmax). Type-II plaques are characterized by equal arc-lengths in the upstream and downstream vascular walls at the position of WTmax. Type-III plaques are characterized by a greater arc-length located in the upstream vascular wall below the position of WTmax. Compared with the other two types of plaques, type-I plaques are more vulnerable to IPH, which may be attributed to the higher shear stress caused by the slope toward the upstream of type-I plaques [137]. A previous study also proved that higher maximum shear stress correlates with calcification and IPH [172].

The “Crouse Score” quantifies the arteriosclerosis degree, defining IMT ≥ 1.1 mm as plaques. The carotid plaque score (CPS) can be calculated as the sum of maximal thickness of each single plaque on the bilateral carotid arteries [173]. The CPS has been proven to be an independent marker of coronary events [174]. Compared with stable plaques, vulnerable plaques exhibited a remarkably higher average CPS, which can be obtained via carotid ultrasound through an easy procedure that has a low cost and involves no radiation [2]. 

### 3.9. Others 

The dietary inflammatory index (DII) represents the total inflammatory effect obtained from food. First, each food or nutrient is divided into proinflammatory or anti-inflammatory foods. Then, a value is assigned for each one based on the influence they exert on serum inflammatory markers. The DII is the sum of the values of all foods or nutrients [175]. High DII scores suggest a severe systemic inflammatory status. The DII correlates with the characteristics of plaque vulnerability, as the higher the DII is, the higher the plaque vulnerability. For patients with ischemic stroke, the DII may be a predictor for carotid plaque vulnerability. Further studies are needed to explore the potential mechanism [4]. 

Pharmacokinetic modeling of dynamic contrast-enhanced magnetic resonance imaging (DCE MRI) can help to noninvasively quantify plaque microvasculature. The volume transfer coefficient K trans reflects microvascular flow, density, and permeability. In addition, K trans is related to intraplaque macrophage accumulation, loose ECM and IPH [176]. Compared with the asymptomatic side, a lower K trans was observed in the entire vessel wall of the symptomatic side, suggesting an abnormal microvasculature caused by extensive necrotic tissue. The higher K trans of the asymptomatic side seems to be irrelevant to slight stenosis or the low incidence of thin fibrous caps and IPH [177]. Macroscopic factors that potentially predict vulnerable plaques are listed in Table 3.

## 4. Conclusions

At the microscopic level, the underlying mechanisms of vulnerable plaques remain unclear. Further studies are needed to detect more potential molecular biomarkers and therapeutic targets. At the macroscopic level, serval imaging technologies currently have the capacity to identify classical features of vulnerable plaques, while some limitations still exist. Combined with potential microscopic mechanisms, developing novel detection technology, grasping the strengths and limitations of technologies and combining the appropriate technologies in necessity are conducive to the early identification and intervention of vulnerable plaques, thereby decreasing the subsequent risk of cerebrovascular events. 

## Figures and Tables

**Table 1 biomolecules-12-01192-t001:** Microscopic factors that potentially predict vulnerable plaques.

Category	Factors	Expression Level	Related Cells	Source	Related Mechanism	References
Inflammatory markers	CRP; hs-CRP	High	Macrophages	Serum; Plaques	Inflammation	[16]
	IL-6, IL-17A, IL-18, IL -21, IL -23, IL-1β,	High	-	Plaques	Inflammation	[10,17,18,19]
	TNF-α	High	T cells	Serum;	Inflammation	[20]
	IFN-γ	Low	Macrophages	Plaques	Macrophage polarization	[10]
	MIP-1β	High	Macrophages	Plaques	-	[10]
	MCP-1	High	Monocytes	Plaques	Monocyte recruitment	[10]
	YKL-40	High	Macrophages	Serum	Cell migration; Angiogenesis; Tissue remodeling	[2]
	SuPAR	High	ECs; Monocytes; T-lymphocytes;	Plaque; Plasma	-	[21]
	S100A12	High	Monocytes; Neutrophils; Dendritic cells	Plasma	-	[22]
MMP	MMP-9, MMP-1, MMP-2, MMP-7, MMP-8, MMP-12, MMP-14,	High	Macrophages	Plaques	ECM degradation	[16,17,20,24,26,27]
The ADAMTS family	ADAMTS4	High	ECs VSMCs; Macrophages;	Plaques; Serum	ECM degradation	[30]
	ADAMTS7	-	VSMCs	Blood	Proteolytic activity; VSMC migration	[28]
Lipid related factors	LOX-1	High	ECs	Serum	EC dysfunction; ECM degradation	[2]
	Non-HDL-C	High	ECs; Monocytes	Serum	Inflammation	[42]
	Cholesterol crystals	High	Macrophages	Plaques	Inflammation	[45]
	Triglycerides	High	-	Serum	Lipid metabolism	[17]
	HDL-C	Low	-	Serum	Lipid metabolism	[17]
	Lp-PLA2	High	Macrophages	Blood	-	[51]
	FFA	High	ECs	Serum	Lipid metabolism	[52]
Cluster Differentiation antigen	CD163	High	Monocyte; Macrophages	Plaques	Inflammation; Angiogenesis; Lipid metabolism; Vascular permeability	[56,60]
	CD36	High	Macrophages	Plaques	-	[57]
	CD146	High	Macrophages	Plaques	Inflammation	[58]
	CD68	High	Macrophages	Plaques	-	[60]
Chemokines	CXCL16	High	Macrophages	Serum	-	[63]
	CCL19	High	CD3+ T-cell lymphocytes	Plaques	-	[64]
MiRNA	MiR-125a	High	LDL-C	Plaques	LDL-C metabolism	[66]
	MiR-133a,	High	VSMCs	Plaques	MMP-9	[66]
	MiR-145,	High	VSMCs	Plaques	Cholesterol release	[66]
	MiR-221	Low	VSMCs	Serum	Inflammation; Angiogenesis; Cell proliferation	[66]
	MiR-532-3p	-	Macrophages	Plaques	MiR-532-3p-CSF2RA axis	[3]
	MiR-124	High	-	Serum	IL-1β and TNF-α	[70]
	MiR-484	High	-	Blood	Angiogenesis	[8]
	MiR-942	Low	-	Blood	Cell apoptosis	[8]
	MiR-214	Low	ECs	Blood	Cell migration; Cell senescence; Angiogenesis	[8]
Other factors	BIRC6	High	ECs; SMCs; Macrophages	Plaques	Apoptosis; Autophagy	[12]
	Osteoprotegerin	High	-	Serum; Plaques	-	[17,83]
	Osteopontin	High	-	Serum	-	[85]
	Insulin	High	SMCs; Macrophages	Serum	Angiogenesis; Macrophage gathering	[87]
	Fibrinogen	High	-	Serum	Secondary hemostasis	[90]
	Fibrinogen γ′	High	-	Serum	Secondary hemostasis	[90]
	VWF	High	-	Plasma	Platelet adhesion; Platelet aggregation	[90]
	VCAM-1	High	Monocytes	Plaques	Monocyte recruitment	[92]
	CMV	High	ECs; SMCs; Monocytes	Plaques	Apoptosis; Cell proliferation; Inflammation	[20]
	BCLAF1	High	SMCs; Macrophage	Plaques	-	[96]
	IRF5	High	Macrophage	Plaques	Inflammation	[97]
	CTH- H_2_S	Low	SMCs	Plaques	Autophagy	[98]
	GLUL	High	-	Plaques	-	[99]

Abbreviations: CRP, C-reactive protein; hs-CRP, high-sensitivity CRP; TNF-α, tumor necrosis factor-α; IFN-γ, interferon-γ; MIP-1β, macrophage inflammatory protein-1β; MCP-1, monocyte chemoattractant protein-1; suPAR, soluble urokinase-type plasminogen activator receptor; ECs, endothelial cells; S100A12, calgranulin C; MMP, matrix metalloproteinase; ECM, extracellular matrix; ADAMTS, a disintegrin and metalloproteinase with thrombospondin motifs; SMCs, vascular smooth muscle cells; VSMCs, vascular smooth muscle cells; LOX-1, lectin-like oxidized low-density lipoprotein receptor-1; non-HDL-C, nonhigh-density lipoprotein cholesterol; HDL-C, high-density lipoprotein cholesterol; Lp-PLA2, lipoprotein-associated phospholipase A2; FFA, free fatty acid; CXCL16, CXC-chemokine ligand 16; CCL19, chemokine (c-c-motif) ligand 19; LDL-C, low-density lipoprotein cholesterol; CSF2RA, granulocyte-macrophage colony-stimulating factor 2 receptor alpha subunit; BIRC6, baculoviral IAP Repeat Containing 6; VWF, Von Willebrand Factor; VCAM-1, vascular cell adhesion molecule-1; CMV, human cytomegalovirus; BCLAF1: BCL2 [B-cell lymphoma 2]-associated transcription factor 1; IRF5: The transcription factor interferon regulatory factor-5; CTH- H_2_S: cystathionine gamma-lyase-hydrogen sulfide; GLUL: glutamine synthetase.

**Table 2 biomolecules-12-01192-t002:** Identification of vulnerable plaques via imaging method.

Imaging Method	Imaging Features of Vulnerable Plaques	Identified Features with High Accuracy	Advantages	Disadvantages	References
US	lower GSM values; Large plaque area (>95 mm^2^); DWA; Large JBA (>6 mm^2^)	-	Fast; Cheap; Available; Noninvasive; No radiation;	Confusion between lipid and IPH	[17,107,108]
3D US	Low GSM values	-	Quantification; Visualization; Monitor therapeutic effects	-	[112,113,114]
UUI	SWE range of 3–5 m/s	-	-	-	[115]
CEUS	Contrast enhancement with high grade and intensity; Grade 2 IPN;	IPN; Ulceration	Fast; Cheap; Available; High compatibility with implants;	Unreliable severe calcification identification; Confusion between IPH and neovascularization; Contrast agents-induced danger	[101,117,120,126]
MRI	Hyperintense on T1-W and TOF sequences; CMBs on T2-W; Significantly discrepant T1 values	IPH	High accuracy	Long scan time; Low slice resolution; Limited spatial coverage; Possible misregistration between different images	[126,129,130,134,135]
CE-MRA	Accumulation of iron oxide with signal absence; Gadolinium enhancement	Ulcerations; IPN	Better vascular imaging;	Expensive; Gadolinium toxicity	[128,129,141,142,143]
BB-MRI	T1-W high signal	-	Better evaluation of lumen area; total vascular area; wall thickness of vessels; Better contrast between blood and tissue;	Dependence on stable comparison with the tissues	[144,145,146]
MATCH	-	Calcification	Reliable calcification identification	Lower slice resolution; Unreliable thin fibrous caps identification	[135,147]
SNAP	-	Ulceration or stenosis colocalizing with IPH	Reliable identification of ulceration or stenosis colocalizing with IPH;	Unreliable chronic or old hemorrhage identification; Confusion between IPH and lipid pools or loose matrix	[148,149]
PET	High 18F-FDG uptake	Lipid-rich plaques; IPN;	Noninvasive; Reliable identification of glucose metabolism; intraplaque inflammation;	Expensive; Ionizing radiation	[17,26,126,150,151]
CTA	Significant enhancement low densities (<25 HU)	Ulcerations; IPN	Better evaluation of plaque volume; plaque thickness; lumen morphology; Vascular remodeling	Radiation; Confusion between IPH and LRNC; Contrast material-induced side effects	[26,129,155,156,157]
DSA	-	-	Gold standard of diagnosing atherosclerosis	Expensive; Radiation; Invasiveness; Late diagnosis	[2,26,52]
OCT	-	Plaque rupture; Plaque erosion;	High-resolution images; Better evaluation of plaque thickness; lipid accumulation	Invasiveness;	[26,158]

Abbreviations: US, ultrasound; GSM: grayscale median; DWA: discrete white area; JBA: juxtaluminal black (hypoechoic) area; IPH, intraplaque hemorrhage; CEUS, contrast-enhanced ultrasound; 3D US: three-dimensional ultrasound; UUI: ultrafast ultrasound imaging; SWE: shear wave elastography; IPN: intraplaque neovascularization; MRI, magnetic resonance imaging; T1-W: T1-weighted; T2-W: T2-weighted; TOF: Time-Of-Flight; CMBs: cerebral microbleeds; CE-MRA, contrast-enhanced magnetic resonance angiography; BB-MRI, black-blood MRI; MATCH, multi-contrast atherosclerosis characterization; SNAP, simultaneous noncontrast angiography and intraplaque hemorrhage; PET, positron emission tomography; 18F-FDG: 18F-Fluorodeoxyglucose; CTA, computed tomography angiography; HU: hounsfield units; LRNC, lipid-rich necrotic core; DSA, digital subtraction angiography; OCT, optical coherency tomography.

**Table 3 biomolecules-12-01192-t003:** Macroscopic factors that potentially predict vulnerable plaques.

Category	Factors	Related Mechanism	References
Geometry and morphology of artery	ER ratio	Hemodynamics	[137,159,161,162,163,164,165,168]
	Maximum wall thickness	Hemodynamics	[170,171]
	Plaque near bifurcation	IPH	
	Incomplete COW	IPH	
Geometry and morphology of plaque	Type-I plaques	High shear stress	[2,137,172]
	CPS	Plaque thickness	
Others	DII	Inflammation	[4]
	K trans	IPH; Loose ECM; Intraplaque macrophage accumulation	[177]

Abbreviations: ER, expansive remodeling; COW, a circle of Willis; IPH, intraplaque hemorrhage; Type-I plaques, plaques with greater arc-length located in the downstream vascular wall above the position of maximal wall thickness; CPS, carotid plaque score; DII, dietary inflammatory index; K trans, the volume transfer coefficient; ECM, extracellular matrix.

## Data Availability

Not applicable.

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
