# Peer review of "Identification Markers of Carotid Vulnerable Plaques: An Update"

_biomolecules, 2022, doi:10.3390/biom12091192_

Round 1
Reviewer 1 Report
Indentifying vulnerable plaque is of great importance clinically. This paper reviewed the identification of vulnerable plaques at both microscopic and macroscopic level. The paper is well structured and written. I only have a few comments for the authors' consideration.
1) I noticed that the manuscript reviewered the papers mostly publised before 2020. The latest literature should be included.
2) 3D US is also an important imaging tool to identify plaque morphology, and many papers about this have been published. This imaging modality should be covered.
3) This review has summarized different techniques and their limitation and strength. The future direction would be suggested to explictly be expressed.
4) proof reading is suggested due to quite a few typos.
Author Response
Response to Reviewer 1 Comments
Point 1:I noticed that the manuscript reviewed the papers mostly published before 2020. The latest literature should be included.
Response 1: We have reviewed the latest literature that published in 2022, and added the related content into the manuscript, highlighted in red via “Track Changes” function.
Point 2: 3D US is also an important imaging tool to identify plaque morphology, and many papers about this have been published. This imaging modality should be covered.
Response 2:Via literature review, the content related to 3D US has been added in to part 3.1 Ultrasound (US) to enrich the manuscript, highlighted in red via “Track Changes” function.
Point 3: This review has summarized different techniques and their limitation and strength. The future direction would be suggested to explictly be expressed.
Response 3:Combined with different techniques in the manuscript, we have added a comprehensive summary in the ending of Part 3 to explicitly express the future direction.
Point 4: proof reading is suggested due to quite a few typos.
Response 4:We have complete proofreading.

Reviewer 2 Report
This review provides laboratory findings and clinical diagnosis technologies for identifying vulnerable carotid plaques, which is excellent information for preventing ischemic stroke in patients.
The authors stated in the manuscript that plaques could be classified as stable and vulnerable. Table 1 shows microscopic factors potentially predicting vulnerable plaque. These are factors for plaques. Please address the details of some factors specific to vulnerable plaques.
Table 2 shows the identification of vulnerable plaques via the imaging method. Please include vulnerable plaques' diagnosis criteria and accuracy in these imaging methods.
Author Response
Response to Reviewer 2 Comments
Point 1:The authors stated in the manuscript that plaques could be classified as stable and vulnerable. Table 1 shows microscopic factors potentially predicting vulnerable plaque. These are factors for plaques. Please address the details of some factors specific to vulnerable plaques.
Response 1:We have explained the related factors in Table 1 that are factors for plaques and maybe not that specific for vulnerable plaques by adding the following explanation to the manuscript: “In Table 1, some factors related to the mechanism of plaque progression (such as lipid metabolism and inflammation) may play roles in the whole process of plaque progression. Accordingly, they may be detected in all plaques, indeed, they are factors for plaques. While the expression level of these factors in stable plaques and vulnerable plaques show significant difference, which also may be valuable to identify vulnerable plaques. In others words, the existence of them may be not specific to vulnerable plaques, while the significantly differential expression level may be specific to vulnerable plaques, and they may be potential markers of vulnerable plaques.”
Additionally, we have added some microscopic factors potentially predicting vulnerable plaques into Table 1. In the present version, we have changed “trend” into “expression level” to explicitly express the differential expression tendency of these factors in vulnerable plaques to avoid ambiguity.
Point 2: Table 2 shows the identification of vulnerable plaques via the imaging method. Please include vulnerable plaques' diagnosis criteria and accuracy in these imaging methods
Response 2:We have added 2 columns (“Imaging features of vulnerable plaques” and “Identified features with high accuracy”) in Table 2 to include the vulnerable plaques' diagnosis criteria and accuracy in imaging methods.

Round 2
Reviewer 1 Report
Thank the authors for addressing most of my concerns. I do not have other comments.
Author Response
Thank you very much for the previous constructive suggestions for us. Thanks to these valuable suggestions, the manuscript has been improved.
Reviewer 2 Report
The line spaces should be adjusted.
Author Response
We have adjusted the line spaces to make the manuscript in good order.